# Deciphering Personalization: Towards Fine-Grained Explainability in Natural Language for Personalized Image Generation Models

## Abstract

Image generation models are usually personalized in practical uses in order to better meet the individual users' heterogeneous needs, but most personalized models lack explainability about how they are being personalized. Such explainability can be provided via visual features in generated images, but is difficult for human users to understand. Explainability in natural language is a better choice, but the existing approaches to explainability in natural language are limited to be coarse-grained. They are unable to precisely identify the multiple aspects of personalization, as well as the varying levels of personalization in each aspect. To address such limitation, in this paper we present a new technique, namely **FineXL**, towards **Fine**-grained e**X**plainability in natural **L**anguage for personalized image generation models. FineXL can provide natural language descriptions about each distinct aspect of personalization, along with quantitative scores indicating the level of each aspect of personalization. Experiment results show that FineXL can improve the accuracy of explainability by 56%, when different personalization scenarios are applied to multiple types of image generation models.

## 1 Introduction

Open-sourced image generation models [6], including diffusion models [19; 31; 46; 49], generative adversarial networks (GANs) [16; 41] and auto-regressive models [47; 60], have been widely used in various applications, ranging from image stylization [30; 56], artistic content creation [27; 33], to virtual character design [28; 55]. In these applications, to meet the different users' personalized needs, pre-trained models are usually fine-tuned with stylized data to derive personalized models [53; 26; 20; 29]. A large quantity of such personalized models have been made available online[1] for others to use, but most of these models offer limited *explainability* regarding how they are being personalized. More specifically, being different from the current practices in explainable AI that explore the correlation between model's input, output and intermediate features [14; 7], such explainability interprets how an image generation model's output changes after personalization [22], in aspects such as specific subjects or styles. For example, a personalized model could generate more stylized images in different ways (e.g., more vivid, artistic, historic, etc). Details about the fine-tuning process, such as the statistical properties of the training data, could provide useful information about explainability, but are also inadequately documented or even missing in the online repositories of most published models online.

Such explainability is important to real-world usage of those personalized image generation models published online. As shown in Figure 1, many users nowadays do not opt to personalize models by themselves with their own datasets, but instead select the most suitable model from those personalized models available online. In these cases, users rely on clear and interpretable explanations for *model selection*. Without such explanations, users will have to manually prompt each model and

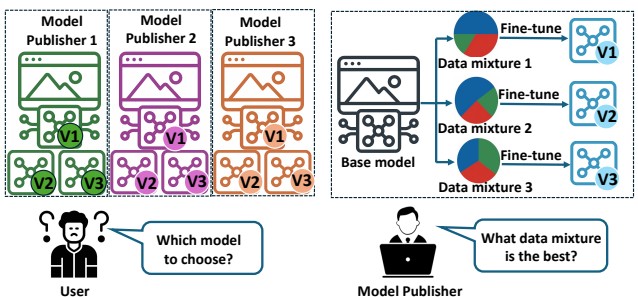

Figure 1: Importance of explainability for personalized image generation models to be used by human users

empirically compare different models' outputs, which is time-consuming or even infeasible considering the large number of available models. On the other hand, even for well-documented and close-sourced image generation models, such explainability could also help the model publisher's

---

[1]On HuggingFace, there are >80,000 personalized text-to-image models made and uploaded by individuals.

fine-tuning process, as it offers insights into each fine-tuned model's outputs and helps adjust the training data if the fine-tuned model's behavior diverges from expectation.

Intuitively, mechanistic AI explainability [5; 15] can be applied to image generation models, and interpret the functionality of different neurons by learning interpretable features using another model, such as a sparse autoencoder [8]. However, these approaches are expensive and are limited to specific types of model structures [59] or labeled data [4; 32]. Alternatively, techniques of visual concept discovery can be used to identify the difference in model's outputs before and after personalization through visual features [21; 63], but it is still difficult for human users to explicitly summarize such differences. As shown in Appendix F, when the model is personalized to generate images with more motion blur, images with more motion blur can be computationally identified as significant [63], but the commonality among these images may not be identified as "more motion blur" in human eyes.

Instead, a better way is to explain the difference of personalized model's output distribution in *natural language*, by leveraging vision-language models (VLMs) for image understanding and interpretation [3]. One can obtain natural language explanations of how models are personalized via image captioning [10; 66], or prompting a VLM to summarize the differences between images generated by the pre-trained model and personalized model [3]. Such summarization can be scaled to an unlimited number of comparisons [23; 71], ensuring that explanation is not biased.

The key limitation of these techniques is that the natural language explanations they provide are *coarse-grained*. In most cases, an image generation model is personalized in multiple aspects, depending on the training data being used. The VLM's summarization, when being used as the explanation of personalized models, is mostly vague and cannot precisely distinguish one aspect from others. For example, when personalization covers both aspects of vividity and abstractionism as shown in Figure 2, the VLM summarization could be "a modern artistic style", from which the two aspects cannot be clearly identified. Similarly, personalization in one aspect could also be done at varying levels [17], which cannot be precisely quantified by the existing techniques.

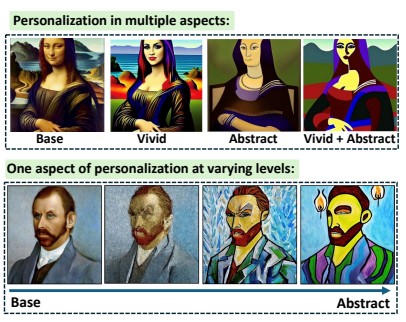

Figure 2: Personalized image generation in multiple aspects with varying levels

To address these limitations, in this paper, we present a new technique, namely **FineXL**, towards **Fine**-grained e**X**plainability in natural **L**anguage for personalized image generation models. More specifically, as shown in Figure 3, we aim to explicitly identify each aspect of how the model is being personalized and provide natural language explanation, along with quantitative scores indicating the level of each aspect of personalization.

To achieve this objective, in the design of FineXL, we first use an image encoder (e.g., the CLIP encoder [44]) to map the divergence between output distributions of the personalized model and the pre-trained model into a high-level representation in a semantically rich representation space, to make the divergence is quantifiable. Then, to further interpret this representation, we leverage a VLM with sufficient representation power (e.g., GPT-4o) to generate multiple low-level natural language concepts, and convert them into vectors in the same representation space. To ensure that these low-level concepts correctly correspond to different aspects of personalization, we only select those concepts whose corresponding vectors are orthogonal in the embedding space. In this way, based on the linear representation hypothesis [61; 35], the high-level representation of the distribution divergence is compositional [57; 62], i.e., vectors of these low-level concepts linearly combine to approximate the overall divergence. The corresponding coefficients of these concepts in the composition, then, serve as scores quantifying the level of personalization in different aspects.

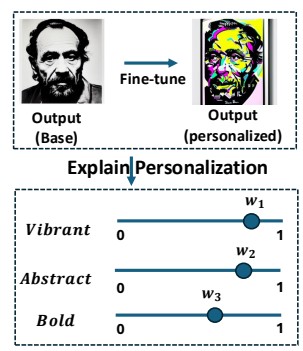

Figure 3: FineXL: fine-grained explanations of a personalized image generation model

We evaluated FineXL over different types of image generation models, including diffusion models, GANs and auto-regressive models, which are personalized using *1)* a synthetic dataset with 400 images in 15 unique styles, *2)* the image Style Transfer dataset [37] with 2,500 images in 50 unique

styles and *3)* the WikiArt dataset [2] with 81,444 images of paintings from 129 artists. From the evaluation results, our main findings are as follows:

- When image generation models are personalized in one aspect with varying levels, FineXL can improve the accuracy of explanation by 56% compared to the baselines.

- In a more challenging scenario where models are personalized in multiple aspects with varying levels, FineXL can reduce explanation error by at least 50% compared to baselines.

- FineXL is completely training-free and generically applicable to all major types of image generation models.

## 2 RELATED WORK

**Explainability of data distributional differences.** Earlier work suggested that the difference between two text datasets can be explained in natural language [69], by fine-tuning a LLM to generate explanations using text data from the two datasets and evaluate their correctness with additional data [68]. This approach was also extended to images [23; 71] and audio [18], by adopting multimodal LLMs. However, these explanations are limited to be coarse-grained and qualitative as shown in Section 1, and such limitation motivates us to further enhance the explainability to be fine-grained.

**Interpreting representations in the embedding space.** Well-trained multimodal foundation models can transform input data into representations within a rich semantic space [13; 65; 54]. For example, an LLM can project tokens into its embedding space [35]. However, such representations are not inherently interpretable. Concept-based interpretability methods propose that a high-level representation (e.g., a green house) consists of multiple low-level concepts (e.g., green and house) [36; 70; 57], and the linear representation hypothesis [43; 45] further suggests that a high-level representation in the model's embedding space can be expressed as a linear combination of vectors corresponding to these low-level concepts. Such correspondence, then, highlights the possibility of achieving explainability from such embedding representation if the low-level concepts are known in advance, and motivates our design of FineXL that linearly decomposes the such representation into a set of low-level concepts, which correspond to orthogonal vectors in the same embedding space.

## 3 PROBLEM FORMULATION

Let $G$ denote an image generation model that generates image $x \in \mathcal{X}$ conditioned on a text prompt $t \in \mathcal{T}$, the model's output distribution can be described as:

$$x \sim p(x|t) = G(t, z), \tag{1}$$

where $z$ indicates random noise. Then, given a pre-trained model ($G_{base}$) and a personalized model ($G_{personal}$), our objective is to quantify the divergence between these two models' output distributions, i.e., $Div[p_{personal}(x|t)||p_{base}(x|t)]$, and explain such divergence using a set of low-level concepts $\mathcal{C} = \{C_1, C_2, \ldots, C_N\}$ in natural language and their associated scores. In practice, a sufficient number of concepts will be involved to ensure that the divergence of output distributions can be precisely explained. More details of deciding the value of $N$ can be found in Section 4.5.

To ensure that each of these concepts represents an aspect of model personalization, we will need to map these concepts into the same representation space as $Div[p_{personal}(x|t)||p_{base}(x|t)]$ using a mapping function $f : C \rightarrow V$, which should also ensure the linear additivity among the mapped concepts, i.e., for any $C_i, C_j \in \mathcal{C}$, there exist $w_i$ and $w_j$, such that

$$f(C_i \cup C_j) = w_i f(C_i) + w_j f(C_j). \tag{2}$$

We have fine-grained explanation of $Div[p_{personal}(x|t)||p_{base}(x|t)]$ via decomposition in Eq. (2):

$$Div[p_{personal}(x|t)||p_{base}(x|t)] = \sum\nolimits_{i=1}^{N} w_i f(C_i), \tag{3}$$

where $w_i$ is the score that indicates how much the model is personalized in the aspect of $C_i$.

## 4 METHOD

In this section, we present our FineXL design, as illstrated in Figure 4. First, we use an image encoder to quantify the divergence between the output distributions of the pre-trained model and personalized model into a high-level representation (Section 4.1). Afterwards, a VLM and a text encoder are utilized to automatically discover a set of orthogonal low-level concepts in natural language related to personalization (Section 4.2–4.4), which are then used to interpret the high-level representation by decomposing it into a linear combination of these low-level concepts (Section 4.5).

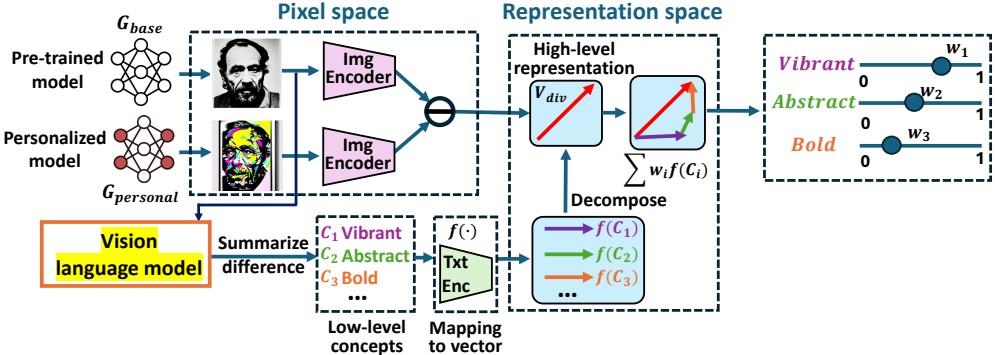

Figure 4: Our design of FineXL: the divergence between pre-trained and personalized models' output distributions is first converted into a high-level representation, which is then linearly decomposed into a set of low-level concepts in natural language about personalization that are suggested by a VLM. More details about this design and each step can be found in Algorithm 1.

## 4.1 QUANTIFYING THE DISTRIBUTIONAL DIVERGENCE

A simple way to evaluate the divergence between two data distributions is to compute their distances in data space. For instance, images can be converted into color histograms and compared using the Earth Mover's Distance (EMD) [52]. However, this captures only low-level features like color [50; 51] and fails to represent styles or complex structural patterns.

Instead, in FineXL, our approach is to utilize an image encoder ($\mathbf{Enc^{img}}$) to extract such divergence into a vector representation ($\mathbf{V}_{div}$), such that

$$V_{div} = \mathbb{E}_{t \in \mathcal{T}}\{\mathbf{Enc^{img}}[G_{personal}(t, z)] - \mathbf{Enc^{img}}[G_{base}(t, z)]\}. \tag{4}$$

In practice, FineXL can adopt any well-trained image encoder for such extraction. However, since in later stages of FineXL we need to further convert the text descriptions of low-level concepts into the image encoder's representation space (see Section 4.3), it is better for the image encoder to share an aligned representation space with a corresponding text encoder. For example, the image encoder from text-image alignment models, such as CLIP [44] and ALIGN [34], is a good choice.

To practically calculate $V_{div}$, we estimate the expectation in Eq. (4) by sampling the corpus of input prompt texts, so that for $n$ text samples $t_1, t_2, ...t_n$ (the number of text samples needed can be found in Appendix E.), we have

$$V_{div} = \sum_{i=1}^{n}\{\mathbf{Enc^{img}}[G_{personal}(t_i, z)] - \mathbf{Enc^{img}}[G_{base}(t_i, z)]\}. \tag{5}$$

## 4.2 DISCOVERY OF LOW-LEVEL CONCEPTS

To automatically discover a set of low-level concepts ($\mathcal{C}$) in natural language related to the divergence of models' output distributions, FineXL utilizes a VLM to summarize the divergence with different pairs of generated outputs from the pre-trained model and personalized model. To ensure sufficient explainability, the VLM will generate summaries multiple times using different pairs of images and take the union of summarized concepts as $\mathcal{C}$. Besides, since each concept

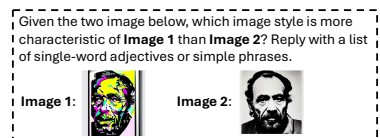

Figure 5: The prompt template for the VLM to summarize divergence of model's output distributions

should correspond to only one aspect of the divergence, we constrain the VLM's summary to a list of single-word adjectives or simple phrases, by explicitly adding instruction into the prompt for VLM, as shown in Figure 5. Sensitivity analysis on the template can be found in Appendix G.

## 4.3 CONVERTING LOW-LEVEL CONCEPTS TO REPRESENTATION SPACE

In order to interpret the distributional divergence ($\mathbf{V}_{div}$) extracted in Section 4.1 using the set of low-level concepts, these concepts in natural language will have to be converted into the same representation space as $\mathbf{V}_{div}$. To do this, we use a text encoder, which is aligned with the image encoder used in Section 4.1, to serve as the mapping function $f$ that maps the low-level concepts to a corresponding set of vectors in this embedding space.

Instead of directly using low-level concepts in natural language as the input to the text encoder, a better approach is to represent each concept as the distributional divergence between two sets of plain texts, being similar to divergence of two image data distributions described in Section 4.1, such that

$$f(\mathcal{C}) = \sum_{i=1}^{n} \{\mathbf{Enc^{text}}[t_i \cup \mathcal{C}] - \mathbf{Enc^{text}}[t_i]\}, \tag{6}$$

where $t_1, ... t_n$ are the set of sampled prompts in Section 4.1.

To ensure that these vectors $f(\mathcal{C})$ can be used to linearly decompose $\mathbf{V}_{div}$, we need to ensure that the representation space of these vectors is *aligned* with the space of $\mathbf{V}_{div}$ and also follows the *linear* representation hypothesis. In the following, we describe our methods of verifying these two properties, and evaluation results about these two properties are in Section 6.7.

**Alignment.** To evaluate the alignment between the two representation spaces, the ground truth of such conversion of low-level concepts needed to be computed with an ideal model ($G_{ideal}$) whose outputs are perfectly aligned with the inputs. If the two spaces are well-aligned, then $f(\mathcal{C})$ should be similar to that computed with $G_{ideal}$, which is

$$\mathbb{E}_{t \in \mathcal{T}} \{\mathbf{Enc^{img}}[G_{ideal}(t \cup \mathcal{C}, z)] - \mathbf{Enc^{img}}[G_{ideal}(t, z)]\}. \tag{7}$$

In practice, we estimate Eq. (7) using a text-to-image model with sufficient representational capacity, such as Stable Diffusion 3.5, and then employ it to select the encoders that best align with our method. The representational capacity of Stable Diffusion 3.5 and the corresponding selection results are presented in Appendix A and Section 6.7, respectively.

**Linearity.** As stated in Eq. (2), if the representation space follows the linear representation hypothesis, the concept vectors should be linearly additive. Therefore, we can verify the linearity using the difference between $f(C_i \cup C_j)$ and $w_i f(C_i) + w_j f(C_j)$. More specifically, each concept $(C_i, C_j)$ is represented by a natural language description, allowing us to form a union of concepts $(C_i \cup C_j)$ by simply combining their descriptions. For example, if $C_i$ corresponds to "vibrant" and $C_j$ to "abstract", then the union of $C_i$ and $C_j$ becomes "vibrant and abstract". As for the coefficients $w_i$ and $w_j$, we optimize them to minimize the difference between $f(C_i \cup C_j)$ and $w_i f(C_i) + w_j f(C_j)$, thereby verifying linearity through the smallest possible discrepancy.

## 4.4 ORTHOGONALITY OF CONCEPTS

Many concepts discovered in $\mathcal{C}$ may have similar meanings (e.g., "vivid" and "vibrant"). To ensure effective and unambiguous explainability, we will exclude concepts that are redundant with each other but only retain the ones with distinct meanings. In particular, existing works showed that in a representation space that satisfies the linear representation hypothesis, concept vectors with distinct meaning should be orthogonal to each other [43]. Therefore, we can constrain the concepts with

$$f(C_i) \perp f(C_j), \forall C_i, C_j \in \mathcal{C}. \tag{8}$$

However, enforcing perfect orthogonality among all the discovered concept vectors may be impractical. Instead, we assess a concept vector's orthogonality by measuring its projections into other vectors. If the total projection exceeds a certain threshold, we consider this concept as redundant. The orthogonality of concept $C_i$ can be computed as

$$orthogonality(C_i) = \sum_{j \neq i, C_j \in \mathcal{C}} \frac{\langle f(C_i), f(C_j) \rangle}{\|f(C_i)\| \|f(C_j)\|}, \tag{9}$$

where $\langle f(C_i), f(C_j) \rangle$ indicates the inner product between $f(C_i)$ and $f(C_j)$.

## 4.5 DECOMPOSING DISTRIBUTIONAL DIVERGENCE

As the final step, we decompose $\mathbf{V}_{div}$ as a linear combination of the retained concept vectors:

$$\mathbf{V}_{div} = \sum_{i=1}^{N} w_i \cdot f(C_i). \tag{10}$$

In practical cases, such decomposition could usually be imperfect due to the error in representation estimation [24]. Instead, we constraint the residual to be smaller than a threshold ($e_{decomp}$):

$$|\mathbf{V}_{div} - \sum_{i=1}^{N} w_i f(C_i)| < e_{decomp}, \tag{11}$$

and more concepts will be involved until such decomposition residual is smaller than $e_{decomp}$. Ablation studies about this threshold can be found in Appendix B.

Based on steps described in Section 4.1-4.5, our design of FineXL is described by Algorithm 1.

---

**Algorithm 1** Fine-grained eXplainability in natural Language (FineXL)

---

**Require:** Base model $G_{base}$, Personalized model $G_{personal}$, a set of probing text prompts $\mathcal{T}$, threshold for identifying concept vector orthogonality $e_{ortho}$ and threshold for decompotistion error $e_{decomp}$

$V_{div} \leftarrow \mathbf{0}, \mathcal{C} \leftarrow \emptyset$    // Compute distribution divergence and discover low-level concepts

**for** $t_i \in \mathcal{T}$ **do**

    $I_{personal} \leftarrow G_{personal}(t_i, z)$ , $I_{base} \leftarrow G_{base}(t_i, z)$

    $V_{div} \leftarrow V_{div} + \mathbf{Enc^{img}}[I_{personal}] - \mathbf{Enc^{img}}[I_{base}]$

    Prompt VLM to propose a subset of low-level concepts $\mathcal{C}_{sub}$ given $(I_{personal}, I_{base})$

    $\mathcal{C} \leftarrow \mathcal{C} \cup \mathcal{C}_{sub}$

**end for**

$V_{div} \leftarrow V_{div}/|\mathcal{T}|$    // Low-level concepts conversion and distribution divergence decomposition

$\mathcal{C}_{ortho} \leftarrow \emptyset$    // The set of orthogonal concepts

**for** $C_i = \mathcal{C}$ **do**

    $f(C_i) = \sum_{t_j \in \mathcal{T}}\{\mathbf{Enc^{text}}[t_j \cup C_i] - \mathbf{Enc^{text}}[t_j]\}$

    $orthogonality(C_i) = \sum_{j=1}^{|\mathcal{C}_{ortho}|} \frac{\langle f(C_i), f(C_j^{ortho})\rangle}{\|f(C_i)\|\|f(C_j^{ortho})\|}$

    **if** $orthogonality(C_i) < e_{ortho}$ **then**

        $\mathcal{C}_{ortho} \leftarrow C_i \cup \mathcal{C}_{ortho}$

        $e \leftarrow \min \|V_{div} - \sum_{i=1}^{|\mathcal{C}_{ortho}|} w_j f(C_j^{ortho})\|$

        **if** $e < e_{decomp}$ **then**

            break

        **end if**

    **end if**

**end for**

**Return** $\mathcal{C}_{ortho}, w_i, ...w_{|\mathcal{C}_{ortho}|}$

---

## 5 EXPERIMENT SETTINGS

**Image generation models.** Our experiments primarily focus on personalization of diffusion-based image generation models, which are the most commonly used models for image generation nowadays. Specifically, our experiments fine-tune the U-Net structures in the Stable Diffusion v2.1 model [48] using the Adam optimizer with a learning rate of 1e-4. Besides, to demonstate the generalizability of FineXL, our experiments also involved other types of image generation models, including ControlGAN [38] and Anole-7B [11] (an auto-regressive image generation model): we fine-tune all the parameters of ControlGAN and only fine-tune the image detokenizer of Anole-7B.

**Datasets.** We mainly use image datasets for art style personalization because it is the most common personalization and results can be easily visualized. We first generate a synthetic dataset with diverse image styles and the ground truth explanations to verify the correctness of FineXL's explanations. Then, we use two real-world datasets (image Style Transfer dataset [37] and WikiArt dataset [2]) to assess FineXL's generalizability. More details about these three datasets are in Appendix D.

**Baselines.** Our baselines, as listed below, include a naive method of explainability and 4 existing approaches on explaining images in natural language. We did not include those using visual features as explanations [21; 63; 23], as such explanations are difficult to be interpreted in natural language.

- **Naive method** asks GPT-4o [3] to summarize the difference into aspects with scores.
- **Diff Caption [66]** is a training-based scheme for captioning difference between images via contrastive learning.
- **Chg2Cap [10]** uses a CNN to extract image features and fine-tunes an encoder-decoder transformer to convert the relationship between image embeddings into text description.
- **VisDiff [23]** identifies differences in images using natural language through a two-stage approach: candidate description generation and re-ranking.
- **GSCLIP [71]** provides a variety of coherent shift explanations in natural language and quantitatively evaluate them at scale.

**Evaluation methods and metrics.** Since no benchmark exists for fine-grained explanations with quantitative measures, we develop an evaluation metric by simulating personalized model selection and measuring selection error. We evaluate two scenarios: (1) personalization along a single aspect

| (a) Synthetic dataset | (b) Style transfer dataset | (c) WikiArt dataset |

Figure 6: FineXL's fine-grained explanations when models are personalized in one aspect with varying levels (low-level concepts with small scores are omitted)

with varying levels, assessed by mean absolute error (MAE) against ground-truth levels; and (2) personalization across multiple aspects, assessed by model selection accuracy (ACC). For baselines using qualitative text explanations, we use GPT-4o to generate quantitative scores in both scenarios. Details of the setup and metric are provided in Appendix C.

**VLMs and Text/Image Encoders.** Correctness of explanation also depends on VLM and text/image encoders being used. We evaluated the performance of different VLMs, including GPT-4o [3], Llama-3.2-11B-Vision [1] and Qwen2-VL-7B-Instruct [64]. Multiple choices of the aligned encoders, including CLIP [44], ALIGN [34], OpenCLIP [12] and EVA-CLIP [58], are also evaluated.

# 6 EXPERIMENT RESULTS

We first use GPT-4o as the VLM and CLIP as the text/image encoders in FineXL, and compare the explanations of FineXL with baselines (Section 6.1 and Section 6.2). Then, we evaluate how different choices of VLMs and text/image encoders affect FineXL's explanations (Section 6.6 and 6.7). Finally in Section 6.5, we demonstrate FineXL's generality on GAN and auto-regressive models.

## 6.1 EXPLANATIONS TO ONE ASPECT OF PERSONALIZATION

In the first scenario with personalization in one aspect, Figure 6 shows that, when different explanations are generated for personalization, the quantitative scores of personalization computed by FineXL are consistent with personalization levels. Table 1 shows that FineXL can correctly identify the varying levels of personalization, and reduce the explanation error by 56% compared to baselines in both synthetic and real-world datasets.

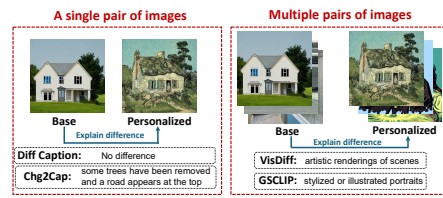

Figure 7: Natural language explanations made by the baseline methods

| Dataset | Synthetic | Style transfer | WikiArt |
|---|---|---|---|
| Naive method | 2.2 | 2.1 | 2.5 |
| Diff Caption | 3.1 | 3.7 | 3.3 |
| Chg2Cap | 3.0 | 3.3 | 2.7 |
| VisDiff | 1.6 | 2.5 | 2.2 |
| GSCLIP | 1.8 | 2.6 | 2.5 |
| **FineXL** | **0.7** | **1.6** | **1.4** |

Table 1: Error (MAE) of identifying the varying levels of personalization in one aspect

| # of personalization levels | 3 | 5 | 8 | 10 |
|---|---|---|---|---|
| Naive method | 0.4 | 1.2 | 1.9 | 2.2 |
| Diff Caption | 0.5 | 1.9 | 2.6 | 3.1 |
| Chg2Cap | 0.5 | 1.6 | 2.4 | 3.0 |
| VisDiff | 0.2 | 0.7 | 1.3 | 1.6 |
| GSCLIP | 0.3 | 0.7 | 1.2 | 1.8 |
| **FineXL** | **0.2** | **0.4** | **0.5** | **0.7** |

Table 2: Errors in explanations with different numbers of levels using the synthetic dataset

Among the baselines, DiffCaption and Chg2Cap perform the worst as they summarize differences based on a single pair of images, which may not capture the overall differences between two distributions. VisDiff and GSCLIP perform better, as they employ a similar approach to summarizing differences from multiple image pairs and selecting the best summarization from multiple trials. However, as shown in Figure 7, due to the ambiguity in natural language, these methods fail in precisely quantifying personalization levels.

We also evaluated the errors in explaining personalization with different numbers of levels in personalization. In Table 2, with more levels in personalization, all methods exhibit higher errors in explanation, but FineXL retain lower errors compared to baselines. Especially when there are many varying levels in personalization that are to be distinguished, FineXL's capability of fine-grained explanation further results in bigger advantages compared to baselines.

## 6.2 EXPLANATIONS TO MULTI-ASPECT PERSONALIZATION

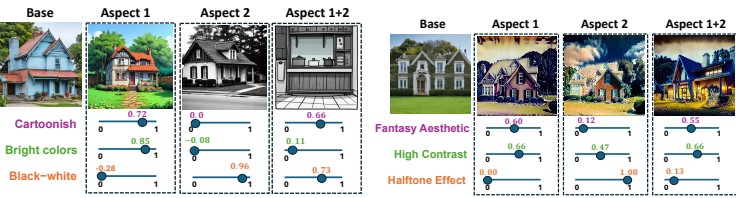

(a) Synthetic dataset        (b) Style transfer dataset

We also conduct experiments in a more challenging scenario where models are personalized in multiple aspects with varying levels on each aspect. As shown in Figure 8, when models are personalized for multiple aspects, the quantitative scores for different

Figure 8: FineXL's fine-grained explanations when models are personalized in multiple aspects

low-level concepts change accordingly compared to the cases where the model is only personalized for one aspect, showing that FineXL can correctly capture the multiple aspects of personalization.

Further, we evaluated the performance of different baseline methods in such scenario, using the evaluation metric in Figure 11. As shown in Table 3, FineXL surpassed the baselines by reducing the estimation error by at least 50%, when there are only 2 aspects of personalization and each aspect contains 3 varying levels of personalization. Based on the conclusion from the previous scenario, FineXL will outperform the baselines even more in more fine-grained scenarios, i.e., with more personalization aspects and more varying levels in each aspect.

| Dataset | Synthetic | Style transfer | WikiArt |
|---|---|---|---|
| Naive method | 72.2% | 61.1% | 66.7% |
| Diff Caption | 55.6% | 38.9% | 44.4% |
| Chg2Cap | 50.0% | 44.4% | 44.4% |
| VisDiff | 77.8% | 88.9% | 83.3% |
| GSCLIP | 77.8% | 83.3% | 72.2% |
| **FineXL** | **94.4**% | **94.4**% | **88.8**% |

Table 3: The performance of FineXL and baseline methods when models are personalized in different aspects with varying levels

### 6.3 EXPLANATIONS TO OTHER TYPES OF PERSONALIZATION

FineXL's framework is adaptable to any form of personalization that can be described by a set of low-level concepts. For instance, FineXL can be extended to explain subject-driven personalization, such as specific facial features in portraits, by modifying the prompt given to the VLM to summarize subject features instead of artistic styles.

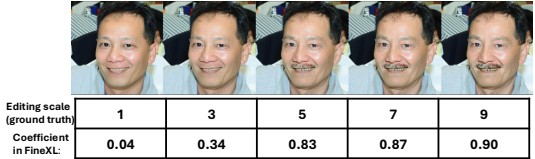

| Editing scale (ground truth) | 1 | 3 | 5 | 7 | 9 |
|---|---|---|---|---|---|
| Coefficient in FineXL: | 0.04 | 0.34 | 0.83 | 0.87 | 0.90 |

Figure 9: Comparing coefficients in FineXL with editing scales as ground truth

We conducted an experiment on the personalization of facial features. To generate images with different levels of personalized images, we utilized the latent space manipulation technique from [17] to generate personalized outputs, where the editing scale serves as a proxy for the ground-truth level of personalization. This approach allows for a controlled evaluation of how FineXL identifies and quantifies changes in a subject's appearance.

As shown in Figure 9, when the personalization aspect is "mustache", the coefficient computed by FineXL correlates positively with the editing scale. It's worth noting that the relationship may not be strictly linear, as the latent space of

| Method | VisDiff | GSCLIP | FineXL |
|---|---|---|---|
| MAE error in Explanation | 0.9 | 0.8 | 0.3 |

Table 4: Error of identifying the varying levels of personalization in one aspect of facial features.

the diffusion model differs from the encoder's embedding space. We also comparing FineXL with two competitive baselines using the experiment setting in Sec 6.1, which is shown in Table 4. These results confirms that FineXL can effectively explain subject-focused personalization.

### 6.4 FINEXL FOR SUBTLE STYLE DIFFERENCE

In sec 6.2, our experiments demonstrate that FineXL can reveal subtle differences between multiple personalized versions of the same model, even when these distinctions are not immediately apparent to the human eye. To further validate FineXL's ability to uncover nuanced distinctions between different image generation models, we conducted experiments comparing Stable Diffusion (SD) v1.4 and SD v2.1, using SD v1.1 as the base model. Our findings in Figure 10, indicate that SD v2.1 tends to generate images with more vibrant and con-

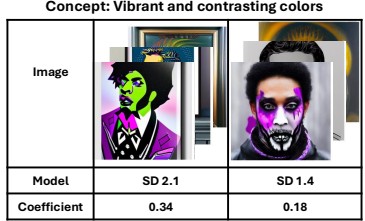

Figure 10: Different SD model versions

trasting colors. This result aligns with the qualitative analysis in [23], confirming that FineXL can systematically identify and quantify known differences between foundational model versions.

### 6.5 GENERALIZABILITY ON DIFFERENT IMAGE GENERATION MODELS

Beyond diffusion models, we also evaluated FineXL on other types of image generation models, including GANs (ControlGAN [38]) and auto-regressive models (Anole-7B [11]), by personalizing these models in one aspect. As shown in Table 5 and 6, for both GANs and auto-regressive models, FineXL exhibits the similar performance of fine-grained explanation with that of diffusion-based image generation models shown in Section 6.1.

| Dataset | Synthetic | Style transfer | WikiArt |
|---|---|---|---|
| Naive method | 2.3 | 2.9 | 2.4 |
| Diff Caption | 2.7 | 2.2 | 2.8 |
| Chg2Cap | 2.5 | 2.3 | 2.6 |
| VisDiff | 1.6 | 1.9 | 2.2 |
| GSCLIP | 1.5 | 2.3 | 2.0 |
| **FineXL** | **0.9** | **1.4** | **1.4** |

Table 5: The performance of Fine-XL and baseline methods on ControlGAN [38]

| Dataset | Synthetic | Style transfer | WikiArt |
|---|---|---|---|
| Naive method | 2.7 | 2.4 | 2.6 |
| Diff Caption | 2.7 | 3.0 | 3.1 |
| Chg2Cap | 2.8 | 2.6 | 3.3 |
| VisDiff | 2.0 | 1.0 | 2.3 |
| GSCLIP | 1.9 | 1.1 | 2.5 |
| **FineXL** | **1.5** | **0.7** | **1.8** |

Table 6: The performance of Fine-XL and baseline methods on Anole-7B [11]

### 6.6 USING DIFFERENT VLMS IN CONCEPT DISCOVERY

We evaluate whether the VLM can accurately propose low-level concepts that capture the patterns in personalized model's output. The synthetic dataset is used, and the style descriptions of its subsets serve as the ground truth for low-level concepts. Ideally, the extracted concepts should be similar to these ground truth descriptions, but it is difficult to quantify such similarity. Instead, we assume that if the identified low-level concepts are accurate, a powerful LLM (e.g., GPT-4o) can match them to the correct ground truth description among a set of alternatives, and the percentage of correct selection then estimate such similarity. As shown in Table 7, we tested 3 different VLMs, and concluded that GPT-4o is the best to be used in FineXL.

| # of alternative descriptions | **5** | **10** | **15** |
|---|---|---|---|
| GPT-4o | 100% | 93.3% | 93.3% |
| Llama-3.2-11B-Vision [1 | 100% | 86.7% | 66.7% |
| Qwen2-VL-7B-Instruct [64 | 93.3% | 80.0% | 80.0% |

Table 7: Correctness of low-level concepts discovered by different VLMs, tested on the synthetic dataset

### 6.7 USING DIFFERENT TEXT/IMAGE ENCODERS

To correctly derive quantitative explanations, FineXL relies on aligned text and image encoders to ensure a shared representation space that follows the linear representation hypothesis. Accordingly, we evaluate multiple text-image alignment models from the following three key features in the representation space:

| Model | Linearity↑ | Orthogonality↓ | Alignment↑ |
|---|---|---|---|
| CLIP [44] | 0.79 | 0.03 | 0.72 |
| ALIGN [34] | 0.81 | 0.05 | 0.65 |
| OpenCLIP [12] | 0.72 | 0.05 | 0.63 |
| EVA-CLIP [58] | 0.73 | 0.02 | 0.69 |

Table 8: Performance of alignment models

- **Alignment between text and image encoders.** If the two encoders are aligned, the vectors computed using Eq. (6) and Eq. (7) should be equivalent. We measure alignment by the cosine similarity between these two results. We use Stable Diffusion V3.5 as $G_{ideal}$ in Eq. (7) and the text prompts are sampled from the Microsoft COCO dataset [42].
- **Linearity** is required to ensure that the high-level representation of distribution divergence $\mathbf{V}_{div}$ can be linearly decomposed into vectors of low-level concepts. We measure such linearity using the cosine similarity between the two $f(C_i \cup C_j)$ and $w_i f(C_i) + w_j f(C_j)$.
- **Orthogonality** among the vectors representing the low-level concepts is necessary to avoid redundant concepts with similar meanings. We assess orthogonality by computing the cosine similarity between vectors of different concepts.

The results in Table 8 show that the performance of different models varies slightly, and we conclude that CLIP has the best overall performance among the 4 choices.

## 7 CONCLUSION

In this paper, we present FineXL, a novel technique providing fine-grained explainability in natural language for personalized image generation models. Comprehensive experiment show that FineXL outperform other natural language explanation methods in fine-grained scenarios.

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

## A  Verification of $G_{ideal}$ and Approximation with Stable Diffusion

We define $G_{ideal}$ as a text-to-image model in which the input text and output images are perfectly aligned. Such a model enables us to test whether Equations (5) and (6) map text and images of the same style into a shared representation space. Specifically, given a textual description of a concept, we check whether the embedding produced by the text encoder is semantically consistent when interpreted through the space of image differences. For example, the embedding $f_{\text{vibrant}}$ should approximate the vector difference between the embeddings of "vibrant" and "non-vibrant" images, i.e.,

$$f(\text{vibrant}) \approx \mathbf{Enc^{img}}(\text{vibrant images}) - \mathbf{Enc^{img}}(\text{non-vibrant images}), \qquad (12)$$

where both sets of images are generated by $G_{ideal}$.

Equation (6) assumes such an ideal model, and the degree of alignment can be evaluated by comparing the difference between Eq. (5) and Eq. (6). In practice, a perfectly aligned model does not exist, so we approximate $G_{ideal}$ with Stable Diffusion 3.5, the best-performing open-source diffusion model available. Although this introduces some misalignment, our experiments show that the error between Eq. (5) and Eq. (6) remains small, implying that the error would be even smaller with a truly ideal model.

Moreover, we hypothesize that a model's representation power directly influences its alignment quality, and thus the accuracy of the error computation. To test this, we evaluated multiple versions of the Stable Diffusion model. Consistent with our hypothesis, newer versions of SD, which generally provide stronger representation power, yield smaller approximation errors.

| The model used as the "ideal" | SD 3.5 Large | SD 2.1 | SD 1.4 |
|---|---|---|---|
| Alignment (cosine similarity between Eq. 5 and Eq. 6) | 0.72 | 0.52 | 0.34 |

Table 9: Alignment results using different Stable Diffusion models as the "ideal" model.

## B  Decomposition Threshold

In theory, we aim to minimize the decomposition error as much as possible. However, in practice, there are always components in the embedding space that cannot be fully decomposed. This means that, regardless of how many low-level concepts are used, the error cannot be reduced to an arbitrarily small value. To prevent FineXL from attempting to extract an infinite number of low-level concepts, we introduce a threshold, i.e., $d_{decomp}$ as a fraction of $V_{div}$ as described in Eq. (10), to limit the decomposition process.

We conducted an ablation study using the synthetic dataset to evaluate the impact of different threshold values in the decomposition process and the final performance of FineXL. As shown in the table below, if the threshold is too small (e.g., 0.02 or 0.05), the decomposition error will never fall below it, meaning that the threshold is not effectively limiting the decomposition process. Such a small threshold will also result in an excessively large number of low-level concepts and hence reduce the effectiveness and representativeness of decomposition itself. On the other hand, if the threshold is too large (e.g., > 0.4), the explanation error, i.e., the error of quantitative explanation measured by the ranking error introduced in Sec 5, increases. Based on our experiments, we empirically set the threshold to 0.2. Notably, even when using a relatively large threshold, the explanation error does not increase significantly, indicating that our method is not highly sensitive to the choice of threshold

| Decomposition Threshold | 0.02 | 0.05 | 0.1 | 0.2 | 0.3 | 0.4 | 0.5 | 0.6 |
|---|---|---|---|---|---|---|---|---|
| Number of Low-Level Concepts Needed | $\infty$ | $\infty$ | 68 | 33 | 19 | 9 | 8 | 5 |
| Decomposition Error | 0.094 | 0.093 | 0.098 | 0.192 | 0.280 | 0.383 | 0.428 | 0.541 |
| Explanation Error | – | – | 0.7 | 0.7 | 0.7 | 0.9 | 0.9 | 1.1 |

Table 10: Performance under different decomposition thresholds.

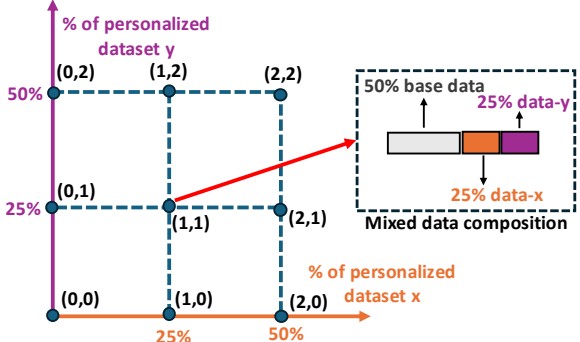

Figure 11: Illustration of mixed training data for personalization. "Based data" refers to data generated by the pre-trained model and represents training data without personalization.

## C  EVALUATION SETUP AND METRICS

We evaluate personalized image generation under two scenarios, each requiring a different evaluation method.

**Scenario 1 (Single-Aspect Personalization).** In this scenario, the model is personalized along a single aspect that varies across 10 levels (e.g., different degrees of vividness from 1 to 10). These levels are implemented by fine-tuning a pre-trained image generation model with different numbers of training steps (e.g., from 20 to 200). Explanations of personalization are evaluated by how accurately the levels can be identified. Specifically, we compute personalization levels by ranking the quantitative values produced by different explanation methods and then compare the rankings with the ground-truth levels using mean absolute error (MAE).

For baseline methods that provide only qualitative (text-based) explanations, we convert the explanations into numerical levels to allow comparison. To ensure fairness, we prompt GPT-4o to assign a score from 1 to 10 based on its interpretation of the text. If the explanation lacks indicators of degree (e.g., "very" or "slightly"), GPT-4o may assign identical or arbitrary values, reflecting that the explanation is too vague to be useful in this scenario. By evaluating rankings rather than absolute values, we ensure that the quantitative outputs from different methods remain comparable even when they are not on the same scale.

In this scenario, we assume that the personalized levels relate to the number of training steps before overfitting, the rationale is: We define personalization level as the distributional divergence between the outputs of the personalized model and those of the base model. At the start of training, the model has not yet adapted to the personalized data, so its outputs remain close to those of the base model, with only faint traces of personalization. As training progresses, repeated fine-tuning updates cause the model to increasingly reflect the personalized data, and its outputs diverge further from the base model. This means that, in general, the degree of personalization increases with the number of training steps. We emphasize that this relationship is not assumed to be linear. Instead, our evaluation only requires that personalization grows monotonically with training steps before convergence—a natural and intuitive assumption, since training for longer exposes the model more strongly to the target aspect or style.

**Scenario 2 (Multi-Aspect Personalization).** In this scenario, the model is personalized across multiple aspects, with varying levels for each. Personalization is achieved by fine-tuning the pre-trained model with training data that mixes different aspects in varying proportions. The proportion of data from a given aspect determines the personalization level for that aspect. For instance, if two aspects are considered (Figure 11) and each has three possible levels, the training data can be combined in six different ways.

Here, explanations are evaluated by how well the computed levels across multiple aspects match the ground-truth mixture. Specifically, we represent the levels of different aspects as coordinates in the space illustrated in Figure 11, and measure evaluation correctness as the accuracy of selecting the true coordinate. For qualitative explanations that lack numerical levels, we again prompt GPT-4o to

infer approximate coordinates from the text description, following the same procedure as in Scenario 1.

## D DETAILS AND EXAMPLE IMAGES OF DATASETS

The synthetic dataset we generate contain 400 images in 15 subsets with unique styles. Each subset is separately generated by prompting the Stable Diffusion 3.5 model [25] with prompts of "object description + image style description". The image style descriptions are manually crafted, while the object descriptions are sampled from the MIcrosoft COCO dataset [42]. A typical example of such prompt could be "a cartoon style photo: A man with a red helmet on a small moped on a dirt road".

The image Style Transfer dataset [37] contains 2,500 images in 50 unique styles. Images in each style are generated by rendering a base content image with a style image, and we use a pre-trained BLIP2 image captioning model [39] to generate the text description. We also use the WikiArt dataset [2] with 81,444 images of paintings from 129 artists. We select 15 artists with the most samples in WikiArt for experiments and consider each artist as a unique image style. In our experiments, we use image style transfer data, but FineXL can apply to any personalization task where image features can be described with keywords or phrases (e.g., facial expressions in human face generation).

Figure 12 and 13 show example images of our synthetic datasets and two real-world datasets, respectively.

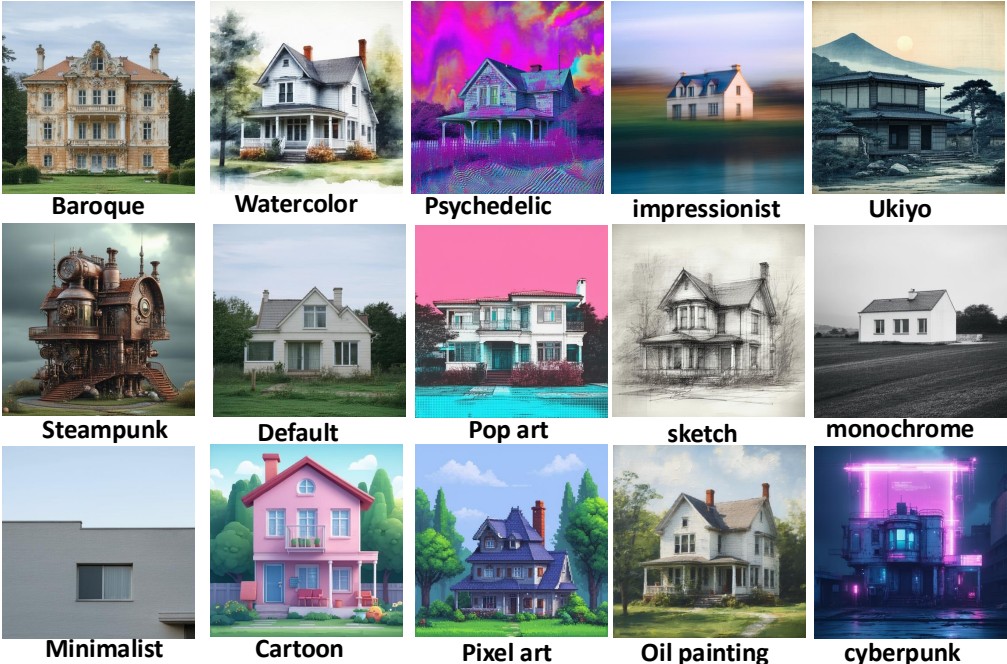

Figure 12: Example images of synthetic data

## E NUMBER OF SAMPLES NEEDED

In Section 4.1 of the main text, to practically calculate $V_{div}$, we estimate the expectation in Eq. (4) by sampling from the corpus of input text samples. Specifically, for n text samples $t_1, t_2, ...t_n$, we conducted experiments to examine how the number of text samples correlates with the error in $V_{div}$'s estimation, measured as the cosine distance from the $V_{div}$ calculated using a sufficiently large volume of samples. Results are shown in Table 11. For the style transfer dataset, since the major differences between images in different styles are primarily in textures, capturing the pattern is relatively easy, requiring only 25–50 samples. For more complex data, such as WikiArt, 100–200 samples are necessary.

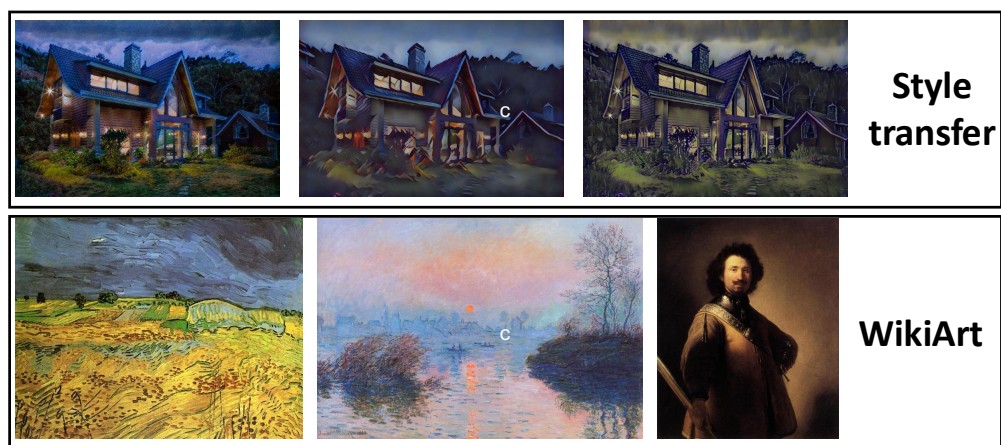

Figure 13: Examples images of two real-world datasets

| # of samples | Style transfer | Wikiart |
|---|---|---|
| 10 | 0.08 | 0.29 |
| 25 | 0.03 | 0.14 |
| 50 | 0.02 | 0.09 |
| 100 | 0.006 | 0.03 |
| 200 | 0.004 | 0.02 |
| 500 | 0.001 | 0.008 |

Table 11: The error of high-level representations with the number of text prompts, measured by cosine distance

## F  VISUAL FEATURE BASED EXPLANATION

Except for natural language based explanation, we can also explain personalization in the form of visual features (63; 40), which may include representative image samples, image patches from the original dataset, or generated features (67; 9). Specifically, the extracted visual features from the personalized model's generated outputs can be compared with those of the base model to explain the corresponding aspects of personalization. However, the extracted visual features are not always easy to interpret and require human users to further explicitly summarize them (23), as shown in Figure 14.

## G  SENSITIVITY TO PROMPT TEMPLATE

We prompt the VLM to discover low-level concepts using a manually crafted prompt template (Figure 5). To assess the robustness of FineXL, we evaluate its sensitivity to different prompt templates. We ask GPT-4o to condense and elaborate the original prompt, and run experiments on synthetic datasets with a single aspect of personalization. As shown in Table 12, the results obtained with different prompts are similar, indicating that FineXL is robust to variations in prompt phrasing when extracting low-level concepts.

| # of personalization levels | 5 | 10 |
|---|---|---|
| Original Prompt | 0.4 | 0.7 |
| Condensed Prompt | 0.5 | 0.7 |
| Elaborated Prompt | 0.4 | 0.7 |

Table 12: Errors in explanations with different prompts with different personalization levels

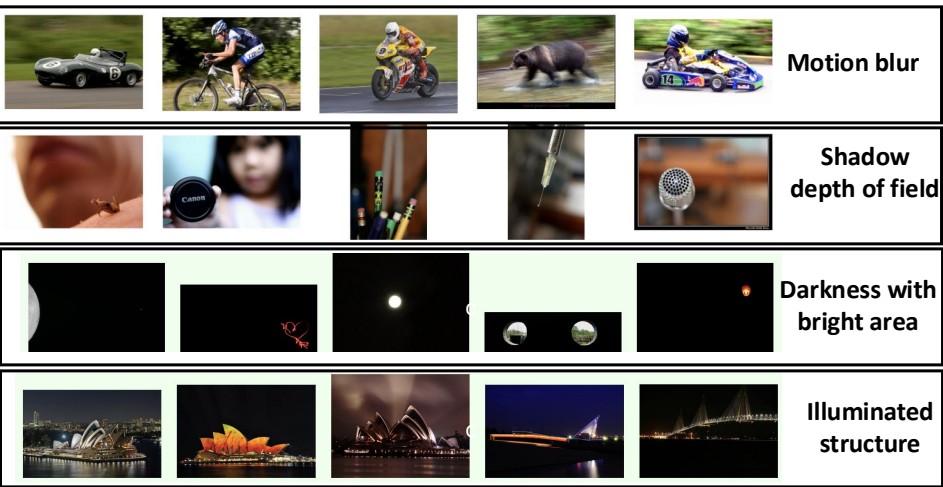

Figure 14: Examples of summarizing image datasets with visual features, the text in the figure is summarized by human.

