# OpenReview forum: "Deciphering Personalization: Towards Fine-Grained Explainability in Natural Language for Personalized Image Generation Models"
_ICLR.cc/2026/Conference — ICLR 2026 Conference Withdrawn Submission_

### Official Review · Reviewer_MPSa · 2025-10-27

**Soundness:** 3
**Presentation:** 2
**Contribution:** 2
**Rating:** 2
**Confidence:** 4

**Summary:**

The paper proposes FineXL, an explainability method designed to characterize the differences between a personalized image generation model and its corresponding base model. FineXL generates paired outputs from the two models and identifies the visual concepts that distinguish them, along with corresponding concept scores. The paper presents supporting results across several personalization scenarios, particularly in cases where multiple aspects of the model are being personalized.

**Strengths:**

- The general direction of the paper, categorizing differences between models, is promising.
- The paper is well written and easy to follow.

**Weaknesses:**

- **Focus on stylistic concepts.** The concepts used to characterize differences between models across generations appear to be mostly stylistic. Visual concepts can exist at different levels of abstraction—such as patterns or textures, color palettes, camera parameters etc. Some of these concepts can be difficult to capture with language, and their decomposition can be learned [1,2]. Additionally, personalized models may encode semantic or cultural biases that was not studied.
- **Categorizing outputs vs. explaining model generations.** The proposed method focuses on categorizing outputs rather than explaining the underlying model behavior. One way to demonstrate the usefulness of the method would be through actionable use cases—for example, using it to characterize a dataset of model weights [3], or to analyze biases and trends in model personalization (e.g., many personalized models being trained for anime characters).
- **More realistic evaluation settings** could strengthen the work. While training personalized models using controlled data is useful for analysis, it would also be valuable to test the approach on existing community models (e.g., from Hugging Face or Civitai). It is also unclear how much value this categorization provides to users when choosing a model. Since multiple image sets are typically generated per prompt, users can often identify their preferred model directly from these outputs without requiring an additional categorization step. A small-scale user study could further assess whether FineXL actually improves users’ ability to choose a preferred model in practice.
- **Clarity of the figures and intended takeaways** could be improved. Figures should provide insight into the method, but some (e.g., Figure 6) have text that is too small to read, and the main message is unclear. The caption suggests that one aspect is being personalized, yet all displayed values vary. It would help to explicitly state which aspect is being personalized and what the figure is meant to convey.

[1] Vinker et al. Concept Decomposition for Visual Exploration and Inspiration.
[2] Gal et al. An image is worth one word: Personalizing text-to-image generation using textual inversion.
[3] Guo et al. ImageGem Dataset.

**Questions:**

- How do the authors train the model to generate data with different levels of personalization of that aspect?
- Did the authors consider semantic differences in personalization e.g., to certain demographics or culture?

---

### Official Review · Reviewer_iAXV · 2025-10-31

**Soundness:** 3
**Presentation:** 3
**Contribution:** 3
**Rating:** 4
**Confidence:** 2

**Summary:**

The paper proposes FineXL, a method for fine-grained, natural-language interpretability of personalized text-to-image models. It maps the output gap between a base model and a personalized model to a distribution-difference vector (V_{\text{div}}) using aligned vision–language encoders; a VLM then proposes a set of low-level concepts, each mapped to the shared embedding space as a concept vector (f(C)) (via text-encoder differences with/without the concept). Under a linear additivity assumption, (V_{\text{div}}) is decomposed onto these concept vectors to yield “concept + strength” scores. On synthetic and real setups, the method reports up to 56% reduction in fine-grained explanation error.

**Strengths:**

Practical objective: Goes beyond “is it personalized” to which aspects changed and by how much, in readable concepts (e.g., vivid/abstract/bold), useful for diagnosis and preference alignment.

End-to-end and controllable: Clear pipeline (concept discovery → concept vectorization → linear decomposition) with orthogonality screening and a residual threshold to curb redundancy and control decomposition depth.

Representation checks: Explicit probes of alignment/linearity/orthogonality across multiple encoders (CLIP/ALIGN/OpenCLIP/EVA-CLIP), with CLIP performing best overall.

Model-agnostic: Demonstrates applicability across diffusion/GAN/AR models and analyzes different VLMs and encoders.

**Weaknesses:**

* **VLM dependency:** Concept discovery relies on a VLM and can be sensitive to the choice of model and prompt templates; the paper could more fully quantify how this propagates to final explanation quality.
* **Strong linear/orthogonality assumptions:** Real concepts are often correlated (e.g., vivid ↔ contrast). Even with orthogonality metrics and thresholds, leakage or non-unique decompositions may occur.
* **Limited stability reporting:** Thresholds for orthogonality (e_{\text{ortho}}) and decomposition residual (e_{\text{decomp}}) are empirically set; more variance reporting across seeds/subsets/encoders would strengthen claims.
* **Output-centric, weakly model-grounded:** Explanations live in CLIP-space projections of images; there’s little attribution to where in the model (layers/blocks/params/attn) the personalization lives or how it causally produces those aspect changes.

**Questions:**

1. Parameter-level attribution. For LoRA/adapter/finetuned weights, report per-module ||ΔW|| (L2 / spectral norm) and rank, and correlate them with concept strengths. Which layers (vision encoder, cross-attn, mid-U-Net) drive “vivid/texture/contrast” concepts?
2. Quantify VLM dependence: Under identical data/encoders, compare different VLMs (and prompt templates) for concept-set Jaccard overlap and concept-weight correlations; attribute variance in final ranking/error to VLM / prompt / encoder choices.
3. Additivity/correlation tests: Beyond current linearity checks, run subspace interventions (remove top-k concepts and re-decompose) and report concept correlation matrices to identify a stable minimal concept set.
4. Sensitivity to the “ideal model”: Replicate alignment evaluation with alternative strong generators; report how alignment scores correlate with final explanation error to rule out circular bias.

---

### Official Review · Reviewer_WFUD · 2025-10-31

**Soundness:** 2
**Presentation:** 3
**Contribution:** 2
**Rating:** 4
**Confidence:** 4

**Summary:**

Authors propose a method FineXL to design a system for personalized models publisher and users to decide which models to publish and which data mixture is the best. The proposed method involves encoding the image and the generated low-level prompts into the same space and then linearly decomposing the image vector.

**Strengths:**

- The idea designed for how can publisher and user to select their own model has value for real-world application.
- Linearly decomposition of image representation to form some "keyword" for a personalization model is interesting.

**Weaknesses:**

- The scope of experiments and design is limited. If the scope is only limited to style personalized, it's hard to cover many general cases in choosing personalized models like subject-driven personalization and abstract concept personalization.
- Lack of elaboration and implementation for claim in figure 1. Author claim that this system can help model publisher to choose the data mixture, but it's missing in this paper. Could author clarify about this part?
- Concerns about the necessity: The authors manually created the art style descriptions when constructing the generated dataset. Therefore, at least in this dataset, there are no cases described by the same style but with slightly different features (this is often where choosing a better model is difficult). Is a direct style description less efficient than such a decomposition? If so, why?

**Questions:**

- How authors decide the prompts for generating images with the personalized model? Are they formatted as "prompt in <style>" or sampled by the VLM?
- Figure 10 is not cited by any section, could author explain the meaning of the numbers in the table? Does it mean that the number of low-level concepts is 33 by "set the threshold to 0.2"

---

### Official Review · Reviewer_32Qq · 2025-11-01

**Soundness:** 3
**Presentation:** 3
**Contribution:** 3
**Rating:** 4
**Confidence:** 2

**Summary:**

This paper addresses the core problem of insufficient explainability in personalized image generation models: existing natural language explanation methods are mostly coarse-grained and cannot accurately identify multiple dimensions of personalization and the degree of personalization in each dimension. To this end, the authors propose FineXL technology, aiming to provide fine-grained natural language explainability for personalized image generation models. Its core workflow is as follows: first, quantify the output distribution difference between the base model and the personalized model through an image encoder to obtain high-level representations; then use a vision-language model (VLM) to discover low-level natural language concepts and map them to the same representation space; after removing redundancy through concept orthogonalization, linearly decompose the high-level representations into a combination of low-level concepts, and use the coefficients corresponding to the concepts as quantitative scores for the degree of personalization. Experiments show that in single-dimensional and multi-dimensional personalization scenarios, FineXL improves explanation accuracy by 56% and reduces explanation error by at least 50% respectively, and is applicable to various image generation architectures such as diffusion models, GANs, and autoregressive models.

**Strengths:**

1. Accurate problem positioning close to practical needs: Focus on the core user demand for "accurate explanation" when selecting personalized models, with experimental design close to real application scenarios (e.g., multi-dimensional personalization, adaptation to different model architectures).

2. Outstanding method generality and practicality: No additional training is required, and it can be directly adapted to mainstream image generation architectures such as diffusion models, GANs, and autoregressive models; it only relies on existing VLMs and encoders (e.g., GPT-4o, CLIP), with low deployment costs and easy promotion.

3. Comprehensive and rigorous experimental design: Covers different datasets (synthetic + real), different personalization scenarios (single-dimensional + multi-dimensional), and different model architectures. The effectiveness of the method is fully verified through comparative experiments, ablation experiments, and sensitivity analysis, with highly credible results.

**Weaknesses:**

1. Insufficient exploration of the limitations of the linear representation assumption: The paper assumes that high-level distribution differences can be linearly decomposed into combinations of low-level concepts, but does not verify the applicability of this assumption in complex scenarios - for example, when there are interaction effects between personalization dimensions or concepts are high-dimensional non-linear features, whether the error of linear decomposition will increase significantly, lacking quantitative analysis and comparison.

2. Insufficient rationality and robustness of concept orthogonalization: Redundancy is only judged by the "projection sum threshold", without considering concept redundancy with similar semantics but non-orthogonal vectors; the threshold setting relies on experimental experience (e.g., 0.2), lacking theoretical basis, and the influence of different thresholds on explanation results is not analyzed.

3. Insufficient completeness of concept discovery: Low-level concepts generated by VLMs rely on summarizing differences between image pairs, and it is not verified whether there are cases of "missing key personalization dimensions"; the scalability of the concept vocabulary is not explored, and it cannot be determined whether it can cover all potential personalization scenarios.

**Questions:**

1. I am very curious about whether the method can be transfer to the recent Unified Models personalization tasks, such as UniCTokens[1]. If you can discuss and do some experiments, I will consider raise my score.

2. In the linear representation assumption, how will the error of linear decomposition change when there are interaction effects between personalization dimensions? Can you provide relevant explanations?

3. What is the basis for selecting the threshold for concept orthogonalization? If the threshold is adjusted to 0.1 or 0.3, how will the explanation accuracy and number of concepts change? Is there a theoretical calculation method for the optimal threshold?

4. In low-resource personalization scenarios (e.g., only 5 training samples), will the quantification accuracy of FineXL for distribution differences decrease? Can you provide a comparison of explanation performance under different sample sizes?

[1] UniCTokens: Boosting Personalized Understanding and Generation via Unified Concept Tokens, https://arxiv.org/abs/2505.14671

---

### Note · Authors · 2025-12-03

I have read and agree with the venue's withdrawal policy on behalf of myself and my co-authors.